# *Candida albicans* Biofilm Inhibition by Two *Vaccinium macrocarpon* (Cranberry) Urinary Metabolites: 5-(3′,4′-DihydroxyPhenyl)-γ-Valerolactone and 4-Hydroxybenzoic Acid

**DOI:** 10.3390/microorganisms9071492

**Published:** 2021-07-13

**Authors:** Emerenziana Ottaviano, Giovanna Baron, Laura Fumagalli, Jessica Leite, Elisa Adele Colombo, Angelica Artasensi, Giancarlo Aldini, Elisa Borghi

**Affiliations:** 1Department of Health Sciences, Università degli Studi di Milano, 20142 Milan, Italy; emerenziana.ottaviano@unimi.it (E.O.); elisaadele.colombo@unimi.it (E.A.C.); 2Department of Pharmaceutical Sciences, Università degli Studi di Milano, 20133 Milan, Italy; giovanna.baron@unimi.it (G.B.); laura.fumagalli@unimi.it (L.F.); jessleitegarcia@gmail.com (J.L.); angelica.artasensi@unimi.it (A.A.); giancarlo.aldini@unimi.it (G.A.)

**Keywords:** *Candida albicans*, cranberry, biofilm, *HWP1*

## Abstract

*Candida* spp. are pathobionts, as they can switch from commensals to pathogens, responsible for a variety of pathological processes. Adhesion to surfaces, morphological switch and biofilm-forming ability are the recognized virulence factors promoting yeast virulence. Sessile lifestyle also favors fungal persistence and antifungal tolerance. In this study, we investigated, in vitro, the efficacy of two urinary cranberry metabolites, 5-(3′,4′-dihydroxy phenyl)-γ-valerolactone (VAL) and 4-hydroxybenzoic acid (4-HBA), in inhibiting *C. albicans* adhesion and biofilm formation. Both the reference strain SC5314 and clinical isolates were used. We evaluated biomass reduction, by confocal microscopy and crystal violet assay, and the possible mechanisms mediating their inhibitory effects. Both VAL and 4-HBA were able to interfere with the yeast adhesion, by modulating the expression of key genes, *HWP1* and *ALS3*. A significant dose-dependent reduction in biofilm biomass and metabolic activity was also recorded. Our data showed that the two cranberry metabolites VAL and 4-HBA could pave the way for drug development, for targeting the very early phases of biofilm formation and for preventing genitourinary *Candida* infections.

## 1. Introduction 

The *Candida* genus is the most common etiological agent of fungal opportunistic infections, ranging from superficial to mucosal and systemic diseases [1]. Infections often arise resulting from a derangement of equilibrium of the commensal yeast and the host [2]. *Candida albicans* is the most common species causing such infections, despite other *Candida* species being on the rise. *C. albicans* is a pleomorphic yeast that is able to filament and form biofilm on a variety of biotic and abiotic surfaces [3,4]. The biofilm architecture and the extracellular matrix released by fungal cells allow the fungus to persist within the host, resulting in recurrent infections.

Besides, the biofilm-organized fungus displays an increased nonspecific tolerance to antifungal treatment, further impairing infection clearance [5].

In the clinical settings, *Candida* urinary infection is a quite common complication of urinary catheterization [6]. If not efficiently treated, such infection can lead to fungus dissemination into the bloodstream, resulting in candidemia [7].

On the other hand, in the community, one of the most common recurrent forms of *Candida* disease is vulvovaginal candidiasis [8]. Filamentation and protease production are common features in this context, and they contribute to local tissue inflammation that, in turn, promotes the persistence of the microorganisms [9]. Biofilm formation might also have a role in promoting *Candida* recalcitrance to antifungal local and systemic treatments [10].

Alternative and complementary strategies to control *Candida* growth, and inhibit its biofilm-forming propensity, could represent an efficient way to eradicate the yeast and to reduce its dissemination from the mucosa, either vaginal or urinary, to adjacent tissues or to the bloodstream.

In a recent paper [11], we demonstrated that cranberry (*Vaccinium macrocarpon*) and its urinary metabolites are able to inhibit *Candida* biofilm formation in vitro, and we identified the most active compounds, which were recovered in the urines of subjects taking cranberry as a dietary supplement, responsible for the observed antifungal activity.

In the present study, we corroborated the activity of cranberry urinary metabolites on clinical strains of *C. albicans*, often behaving differently from reference strains, and we characterized the possible mechanism of action of the most promising metabolites, which were 5-(3′,4′-dihydroxyphenyl)-γ-valerolactone (VAL) and 4-hydroxybenzoic acid (4-HBA).

## 2. Materials and Methods 

### 2.1. Culture Conditions and Standardization

Clinical *C. albicans* isolates used in this work were chosen among strains collected during a previous national survey on yeast infections [12]. In particular, the selected isolates for this study were from the genitourinary tract (8 from vaginal swabs and 2 from urine) infections. The reference strain *C. albicans* SC5314, a strong biofilm producer, has been used as a control strain in all experiments. All the isolates were stored at −80 °C until use.

To induce biofilm formation, strains were thawed and grown overnight in yeast extract peptone dextrose (1% *w*/*v* yeast extract, 2% *w*/*v* peptone, 2% *w*/*v* dextrose—YPD) broth at 30 °C in an orbital shaker. Cells were collected by centrifugation, washed twice with cold phosphate-buffered saline (PBS, pH 7.4) and resuspended in 2 mL of PBS. Yeast cells were then counted using a hemocytometer and adjusted to the desired final concentration by dilution in PBS. The final concentration was also checked by plating the yeast suspension onto Sabouraud dextrose agar supplemented with chloramphenicol and incubating for 24 h for colony-forming unit (CFU) count.

### 2.2. Crystal Violet Assay for Biofilm Quantification

*C. albicans* strains were standardized to 10^6^ yeast cells/mL in Roswell Park Memorial Institute (RPMI)-1640 medium. Crystal violet (CV) assay was carried out as previously described [13] with slight modifications. Briefly, 100 μL of the standardized inoculum was seeded in each well of polystyrene 96-well plates and incubated at 37 °C for 1 h to promote adhesion. Non-adherent cells were then removed, and wells were washed with warm RPMI-1640; eventually, the medium was replaced, and the plate further was incubated for 24 h. At 24 h, total biomass measurement was carried out by crystal violet staining (0.1% *w*/*v*) and the absorbance was measured at 540 nm using an EnSight microplate reader (PerkinElmer). Three independent experiments were carried out with five technical replicates for each condition. The reference strain *C. albicans* SC5314 was used as control. Clinical isolates were classified as high (+++), medium (++) and low (+/−) biofilm producers according to Marcos-Zambrano et al. [14].

To evaluate the inhibitory effect of various cranberry metabolites, we tested both urinary fractions collected post cranberry intake and purified compounds.

Urine fractions were obtained from healthy volunteers taking 2 capsules/day of a highly standardized cranberry extract (Anthocran^®^, Indena S.p.A., Milan, Italy) for one week [11]. Urine, collected at 1, 2, 4, 6, 10, 12, 24 h after the last capsule, was filtered on 0.22 µm filters, lyophilized and solubilized in RPMI-1640 on the day of the assay. A two-fold dilution of each fraction was used in the CV assay.

The following metabolites, detected in urinary fractions by targeted high-resolution mass spectrometry (MS) [11], were purchased from Merck (KGaA, Darmstadt, Germany): 2-hydroxybenzoic acid, 4-hydroxybenzoic acid, 2,3-dihydroxybenzoic acid, 2,5-dihydroxybenzoic acid, and 3-(4-hydroxyphenyl)-propionic acid, as well as protocatechuic acid, kaempferol, quercetin, syringetin from Extrasynthese (Genay Cedex, France). Due to the lack of commercial availability of valerolactone standards, 5-(3′,4′-dihydroxyphenyl)-γ-valerolactone was synthesized in our laboratories [11,15]. For 4-HBA and VAL, various concentrations (3.5–36 µM and 10–245 µM, respectively) were assessed. For the other compounds 10 μM concentration was used. Compounds and urinary fractions were added in the adhesion phase (i.e., at seeding) and then again after the washing step at 1 h.

### 2.3. XTT Reduction Assay

Alterations in the metabolic activity (i.e., cell viability) of yeast cells, treated with different compounds, was evaluated using the 2,3-bis(2-methoxy-4-nitro-5-sulfophenyl)- 2H-tetrazolium-5-carboxanilide inner salt (XTT) reduction assay in a 96-well plate [16]. Briefly, inocula were prepared as above described by inducing biofilm growth overnight in YPD at 37 °C and seeding yeast cells (10^5^ yeast cells/well) in RPMI-1640, as described for CV assay. At 24 h, culture medium was removed, biofilms were carefully washed with PBS, and 100 µL of freshly prepared XTT solution was added to each well. The plate was incubated at 37 °C, in the dark, for 2 h. Colorimetric changes were read with EnSight microplate reader at 450 nm with a reference scanning of 620 nm.

### 2.4. Cell Surface Hydrophobicity Assay

To determine the effect of Anthocran^®^ metabolites on *C. albicans* cell surface hydrophobicity (CSH), the ability of cells to adhere to a hydrocarbon source (Octane, Sigma-Aldrich, Milan, Italy) was measured. Briefly, the reference strain *C. albicans* SC5314 was grown overnight in YPD under orbital shaking in RPMI-1640 alone or supplemented with 0.1 mg/mL Anthocran^®^, 30 µM and 245 µM of VAL, and 4-HBA 35 µM. Cell suspensions were then diluted in PBS to an OD600 of 1, and 1.2 mL transferred to a glass tube. Then, 300 µL of octane was added, the samples were mixed by vortexing for 3 min and the phases were allowed to separate for 10 min at 30 °C. The aqueous phase was then harvested, and its optical density was measured spectrophotometrically. The relative percentage of CSH was calculated as follows: (1)[(OD600 of control−OD600 after octane)÷OD600 of the control]×100

### 2.5. Confocal Laser Scanning Microscopy (CLSM)

To investigate the morphological and structural changes induced by Anthocran^®^ and its metabolites on hyphal growth and biofilm, *C. albicans* reference strain SC5314 was used for CLSM evaluation after 24 h exposure to 30 µM or 245 µM of VAL and 0.1 mg/mL of Anthocran^®^. Biofilms were grown for 24 h on plastic coverslips (Sarstedt S.r.l., Verona, Italy) set at the bottom of a 24-well plate, fixed with methanol 100% and stained with 0.05% calcofluor white (Sigma-Aldrich, Milan, Italy). The thickness of biofilms was measured by CLSM Z-scanning.

### 2.6. CaCo-2 Culture and Cytotoxicity Assay

VAL and 4-HBA cytotoxicity was assessed by 3-(4,5-dimethylthiazol-2-yl)-2,5-diphenyl tetrazolium bromide (MTT) assay on colorectal adenocarcinoma cell line Caco-2 (ATCC^®^ HTB-37™, ATCC, Manassas, VA, USA). Compounds’ stock solutions were prepared in dimethyl sulfoxide (DMSO), diluted in complete medium (Dulbecco’s modified Eagle’s medium supplemented with 10% fetal bovine serum, 2 mM L-glutamine, and 1 mM sodium pyruvate) and filtered on 0.22 µm filters. The concentrations tested were 245, 100, 30, 10 µM for VAL and 36, 18, 3.5, 1 µM for 4-HBA. DMSO 0.1% (the highest concentration in the dilutions) was added as control. Briefly, 1 × 10^4^ Caco-2 cells/well were seeded in a 96-well plate and treated with serial dilutions of VAL and 4-HBA for 24 h. The MTT solution (0.5 mg/mL) was then added to each well and incubated for 3–4 h. Formazan crystals were solubilized with 100 µL/well of lysis buffer (8 mM HCl and 0.5% NP-40 in DMSO) and the absorbance measured at 575 nm in a microplate reader (Power Wave HT, Biotek, Bad Friedrichshall, Germany). The cell viability was calculated by the following formula: (2)% Cell viability=(absorbance sample÷absorbance control)×100

Three independent experiments were performed with four technical replicates for each condition.

### 2.7. Gene Expression Analysis

RNA was isolated from *C. albicans* SC5314 culture at two different time points (4 h, 24 h post biofilm induction in RPMI-1640). RNA extraction was carried out by harvesting adherent yeast cells with 1 mL of TRI Reagent^®^ (Sigma-Aldrich), with few modifications to allow a complete fungal cell wall rupture, and increasing RNA recovery. Briefly, 200 mg of glass beads (106 µm, Sigma-Aldrich) were added to each sample, vigorously mixed in a tissue-lyser (Qiagen, Hilden, Germany) for 3 min at the maximum speed and immediately put in ice to preserve samples from heating. This step was repeated three times. Beads were removed by centrifugation and 200 μL of acid phenol was added to allow the complete fungal cell wall rupture. The aqueous phase was obtained by adding 200 μL chloroform to each sample. After adding 500 μL isopropanol and mixing by vortex, samples were centrifuged to obtain the RNA pellets, which were washed with ethanol 95% and dissolved in 30 μL DNase–RNase-free water. Purity and concentration of RNA extracts were measured by a NanoDrop 2000 spectrophotometer (Thermo Fisher Scientific, Wilmington, DE, USA) and cDNA was performed by the high-capacity RNA-to-cDNA kit (Applied Biosystems, Foster City, CA, USA), according to the manufacturer’s instructions. The reaction mix was prepared using TB Green Premix Ex Taq/ROX qPCR master mix (Takara Bio Europe SAS, Saint-Germain-en-Laye, France), and the following primers: HWP1 (5′-TGCTATCGCTTATTACATGTTATC-3′, and 5′-GAGCTTCTTCTGTTTCACCTTGAC-3′); ALS3 (5′-CAACTTGGGTTATTGAAACAAAAACA-3′, and 5′-AGAAACAGAAACCCAAGAACAACCT-3′); IMH3 (5′-TATTCATATGGCATTATTGGGTGGTA-3′, and 5′-AACCATTTCTGCTTGTTCTTCAGA-3).

The reaction was conducted in a StepOne Plus real-time PCR system thermocycler (Applied Biosystems, Thermo Fisher Scientific, Waltham, MA, USA). Relative expression levels were calculated using the ΔΔCt method using the housekeeping gene IMH3 for normalization. Samples were run in triplicate and three independent experiments were carried out.

### 2.8. Statistical Analysis 

Statistical analyses were performed by GraphPad Prism 7.0e (GraphPad software, La Jolla, CA, USA). For biofilm measurements, the non-parametric Mann–Whitney U test was used for comparing experimental groups. For quantitative RT-PCR experiments, data were statistically analyzed applying a two-tailed *t*-test. In all analyses, *p* < 0.05 was considered statistically significant.

## 3. Results

### 3.1. Inhibition of Biofilm-Formation of C. albicans Clinical Strain by Urine Fractions

Due to the *C. albicans* propensity to form biofilms that are highly strain-dependent, we first checked the selected clinical isolates for biomass production by CV assay. Among the tested isolates, five resulted in strong producers, whereas five were low-biofilm producers (Table 1).

Urine fractions, collected at various time points from healthy donors taking Anthocran^®^, were then tested for their capability to inhibit biofilm formation. As observed in the previous study, on the reference *C. albicans* strain SC5314 [11], the urinary fraction that was collected 12 h after the last pill intake resulted in the higher biomass production inhibition (Figure 1A), together with a reduction in yeast cell viability within the biofilm (Figure 1B). Similar to what was previously reported, the urine collected before Anthocran^®^ supplementation resulted in no significant inhibition of the clinical isolate biofilms when diluted 1:2 in RPMI medium as per the experimental protocol [11]. 

### 3.2. Cranberry Metabolites Activity Against C. albicans Biofilm Formation

MS qualitative analysis, based on a target approach, highlighted several increased metabolites in active urine fractions [11]. We thus tested the direct effect of these metabolites (10 μM) against the *C. albicans* SC5314 biofilm. Among these, 4-HBA resulted as the most active in inhibiting biofilm biomass production (Figure 2). After Anthocran^®^ oral intake, the in vivo concentrations of 4-HBA ranged from 2.51 (± 1.44) μM to 23.84 (± 14.69) μM, reaching 3.71 (± 2.07) μM in the most-active urine fraction (U-12 h).

Untargeted MS analysis of active urine fractions revealed an enrichment in valerolactone/valeric acid derivatives, which is a finding previously unreported in the literature [11]. In particular, the most active fraction (U-12 h) was characterized by the presence of high concentrations (245.24 ± 143.59 μM) of 5-(3′,4′-dihydroxyphenyl)-γ-valerolactone. 

In light of the mass spectrometry results, both targeted and untargeted, we investigated the activity of several concentrations of 4-HBA (Figure 3A), the most active compound from targeted MS analysis, and VAL (Figure 3B) on *C. albicans* clinical isolates, and we demonstrated a dose-dependent inhibition of biofilm formation (CV assay). Further, 4-HBA significantly inhibited the biofilm formation of clinical isolates, only at concentrations equal (3.5 μM) to those observed in the active urinary fraction; indeed, the 18 μM concentration resulted in a bimodal response, inhibiting some isolates and unaffecting others (Figure 3A). VAL activity showed a more pronounced and dose-dependent response, starting from 30 μM (Figure 3B), i.e., the average concentration of VAL recovered in non-active urinary fractions. All the tested doses were maintained in the ranges found in the in vivo study, to verify the activity of the bioavailable metabolites.

To gain insights into biofilm modification by VAL and 4-HBA, in terms of cell morphology, biofilm architecture, and thickness, we analyzed confocal microscopy images of biofilms grown on plastic coverslips (Figure 4). Both 4-HBA and VAL reduced biofilm thickness (about 41% and 63%, respectively). The tested concentrations have been selected according to the results of a dose-dependent assay (the most active in inhibiting biofilm biomass).

### 3.3. 4-HBA and VAL Cytotoxicity on Intestinal Cells

Despite the tested concentrations being close to those identified in the urine, we assessed, in vitro, whether VAL and 4-HBA might exert toxic effects on intestinal cells. We thus applied a cytotoxicity test using Caco-2 cells and the MTT assay. We measured cell viability after treatment with four different concentrations of both 4-HBA and VAL. None of the assessed concentrations resulted in a decrease in cell viability (Figure 5).

### 3.4. 4-HBA and VAL Inhibitory Activity on Candida Adhesion

To elucidate the possible mechanism of the cranberry metabolites inhibitory action, we investigated possible alterations in the cell surface hydrophobicity (CSH). *C. albicans* SC5314 was cultured either in the planktonic or in the sessile (biofilm) form, in the presence/absence of whole cranberry extract (0.1 mg/mL), VAL, and 4-HBA (Figure 6). Biofilm growth resulted in a dramatic increase in CSH compared to planktonic cells (99% and 35%, respectively). While the whole cranberry extract reduced biofilm hydrophobicity (*p* = 0.0042), and VAL and 4-HBA alone did not significantly alter the yeast surface hydrophobicity.

We then evaluated the expression of genes encoding for proteins that are involved in the initial phases of *C. albicans* biofilm formation. In particular, we studied the expression of *HWP1* and *ALS3* genes, which are both involved in adhesion and filamentation. As shown in Figure 7, 4 h after biofilm induction, both genes were downregulated in *C. albicans* SC5314 treated with VAL and with 4-HBA (also by Anthocran^®^ itself), compared to the controls. At 24 h, the downregulation in *HWP1* and *ALS3* gene expression was not observed, which is in line with the reported involvement of both transcripts in the early phases of biofilm formation.

## 4. Discussion

The results reported in the present study ground from a previous work that identified cranberry metabolites, other than proanthocyanidins (PACs), in human urine, upon highly standardized cranberry extract oral intake [11]. The antiadhesive properties of cranberry have long been reported [17,18,19], mainly concerning *Escherichia coli*, which represents the leading uropathogenic agent that is responsible for about 80% of urinary-tract infections (UTIs).

Although the activity of cranberry products is principally attributed to PACs, we were not able to recover, in the analyzed urinary fractions, neither A-type nor B-type PACs. The literature data about PACs excretion in urine are still controversial. Our data are in line with other reports that failed to identify, in human urine, this class of polyphenols, in particular procyanidin A2 [20,21], even if other studies demonstrated that low concentrations of PACs are detectable in urine [22,23]. 

We assessed the activity of cranberry, both crude extract and its urinary metabolites upon oral intake, in reducing *C. albicans* adhesion and biofilm formation on abiotic surfaces (i.e., polystyrene). Indeed, *Candida* spp. are responsible for a high percentage of catheter-related UTIs within the nosocomial setting, which, if not efficiently treated, can eventually lead to bloodstream invasion [24]. *Candida* spp. and, in particular, *C. albicans* are prone to develop biofilms on indwelling devices [25] that offer this fungus new niches to colonize, and to use as an entry route to deeper sites. To reduce the infection burden due to biofilm-colonized medical devices, the use of compounds with anti-adhesive properties to coat or to functionalize biomaterials is also under investigation [26,27]. Among genitourinary infections, *Candida* is also a common etiological agent of vulvovaginitis [28], affecting a high percentage of women at reproductive age. Vulvovaginal candidiasis often develops in a recurrent form of disease that dramatically impacts patient quality of life [28].

*Candida* recalcitrant infections are sustained by strains that are able to switch from a yeast form to a filamented morphology, which helps the fungus in adhesion and immune evasion, and biofilm producers [3]. Because of the high impact of these virulence factors in the pathophysiology and in the outcome of the infection, a therapeutic approach addressing adhesion, filamentation and biofilm formation is strongly desirable.

Cranberry extract, in particular PACs, has been reported to inhibit in vitro *Candida* adhesion, growth, and biofilm formation [29,30]; however, its presence in human urine is still a debate. Hence, we reasoned whether urinary-derived metabolites could retain the observed inhibitory effects. Starting from the most-active urinary fraction in inhibiting *Candida*, i.e., 12 h following the last capsule ingestion, we investigated whether the metabolites peaking in these fractions could account for the anti-biofilm effects per se.

Among several identified metabolites, the following two resulted as the most promising in *C. albicans* virulence modulation: 5-(3′,4′-dihydroxy phenyl)-gamma-valerolactone and 4-hydroxybenzoic acid. Valerolactone derivatives have been already reported by authors investigating urinary cranberry metabolites [31,32]; however, probably due to the highly concentrated amount of PACs in the Anthocran^®^ formulation, urinary fractions at 12 h were particularly enriched in valerolactones.

A recent study demonstrated that valerolactones and their conjugates can counteract *E. coli* [32] adhesion, confirming that the inhibitory activity is not only due to whole PACs, but also to the derived metabolites.

VAL, at the concentration found in urinary fractions, was efficient in reducing *C. albicans* biofilm biomass and thickness, probably affecting the stability of the whole biofilm structure. Many natural compounds [33] have been investigated for their anti-*Candida* effects, but for several of them, the in vitro determined minimal inhibitory concentration (MIC) is beyond the reasonable values, and thus is probably ineffective in vivo. Zida and coworkers [33] suggested that only compounds with MIC values below 1 mg/mL are considered noteworthy for further analysis.

Notably, both the VAL and 4-HBA concentrations tested in our study are way below the suggested cut-off MIC value, as the higher dose assessed was 245 μM (0.05 mg/mL) and 3.5 μM (0.0005 mg/mL), respectively. 

Another issue concerning natural compounds is that the exact mechanism of action is often unknown. To gain some insights into the possible mechanism, we evaluated the following two important features in biofilm formation: modifications of the *Candida* cell wall, i.e., cell surface hydrophobicity, and the expression of genes directly involved in biofilm formation. *Candida* switching from a planktonic to sessile lifestyle has been demonstrated to be sided by important changes in the cell wall [34]. Indeed, fungal cell wall architecture changes dramatically between sessile and planktonic cells [34], and, especially for *C. albicans*, between filamentous and yeast forms [35]. Variations in the carbohydrate and protein composition impact the cell surface hydrophobicity, which, in turn, regulates the initial adhesion to surfaces, in particular to abiotic substrates [36]. Moreover, a positive correlation was observed between hydrophobicity and biofilm biomass [37]. Hence, cell surface hydrophobicity has been suggested as a possible marker for a strain propension to form biofilm [37,38], as it mediates non-specific interactions with the substrate for cell attachment.

We demonstrated that only Anthocran^®^ was able to significantly reduce *C. albicans* cell wall hydrophobicity, suggesting that the observed inhibitory action of VAL and 4-HBA might result from other mechanisms.

From a molecular point of view, surface attachment is mediated by a plethora of surface proteins, such as Als1p, Als3p, Ece1p, and Hwp1p [39]. We thus investigated whether cranberry metabolites could alter the gene expression of *ALS3* and *HWP1*, which are two genes encoding for glycosyl phosphatidylInositol (GPI)-anchored proteins, interacting with each other and promoting the adhesion to abiotic and biotic (host cells) surfaces [40]. At 4 h post biofilm induction, in the presence of VAL and 4-HBA, we observed a significant down modulation of both the *HWP1* and *ALS3* genes. Previous studies demonstrated the crucial role of *HWP1* in the very early phases of biofilm formation, as *HWP1* null mutant results in dramatic changes in the cell wall consistency and thickness [41], and in reduced adherence ability in an in vivo catheter model [40].

Hwp1p also has a key role in *C. albicans* filamentation and adhesion to mucosal surfaces [42]. Although filamentation is not necessary for biofilm formation, as shown for other *Candida* species, such as *C. glabrata*, with a biofilm that lacks the presence of pseudohyphae and/or hyphae [43], filamented forms of *C. albicans* stabilize the three-dimensional structure of the biofilm and promote its resistance to detachment [44]. VAL and 4-HBA inhibitory activity on *ALS3* and *HWP1* was not confirmed at 24 h, which is in line with the precocity of their expression within the biofilm formation process.

Our results suggest that cranberry and/or VAL and 4-HBA could be considered among the preventive approaches for targeting the very early phases of biofilm formation.

Further researches aimed at elucidating VAL and 4-HBA protein target, as well as their possible inhibitory activity against mature *Candida* biofilms, will provide a comprehensive picture on the possible use of VAL and 4-HBA for treating fungal infections.

## Figures and Tables

**Figure 1 microorganisms-09-01492-f001:**
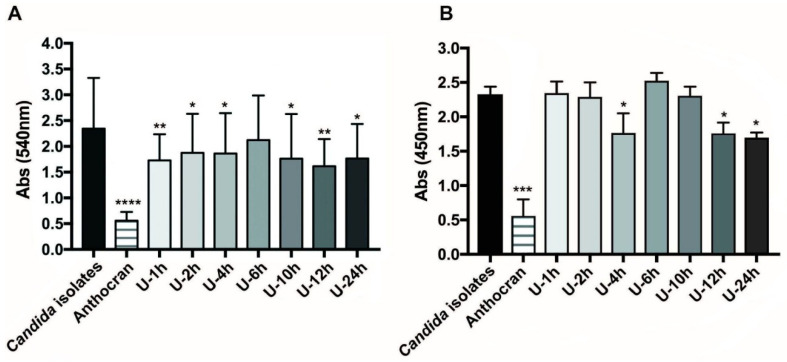
Activity of Anthocran^®^ (0.1 mg/mL), and urinary fractions against ten *C. albicans* clinical isolate biofilms (black bar). Bar graphs show the mean absorbance of all the clinical isolates and error bars are standard deviations. (**A**) Crystal violet assay was used to evaluate biofilm biomass reduction. (**B**) XTT assay was used to measure reduction in biofilm metabolic activity. Data from three independent experiments are reported. Significant differences are indicated by * *p* < 0.05, ** *p* < 0.01, *** *p* < 0.001, **** *p* < 0.0001, Mann–Whitney U test.

**Figure 2 microorganisms-09-01492-f002:**
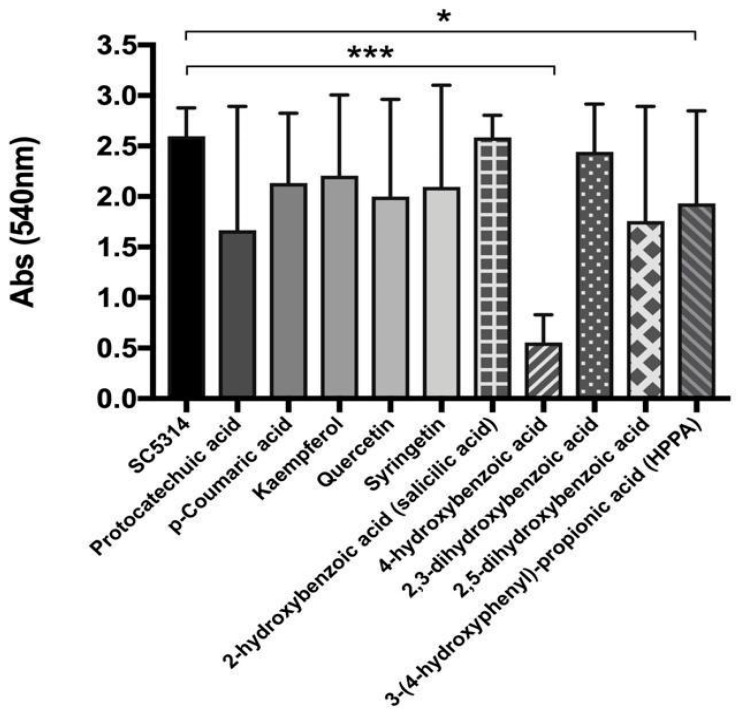
Activity of metabolites (10 μM) identified by targeted MS analysis on *C. albicans* SC5314. Crystal violet assay was used to evaluate biofilm biomass reduction. Bars represent the mean values (± SD) of three independent experiments. Significant differences are indicated by * *p* < 0.05, *** *p* < 0.001, Mann–Whitney U test.

**Figure 3 microorganisms-09-01492-f003:**
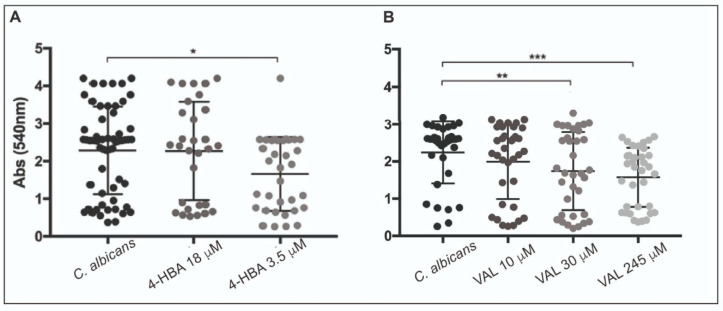
Inhibition of *C. albicans* clinical isolates biofilm formation in the presence of a different concentration of (**A**) 4-hydroxybenzoic acid (4-HBA) and (**B**) 5-(3′,4′-dihydroxyphenyl)-γ-valerolactone (VAL). The results are shown as a scatter plot, where horizontal bars indicate mean and vertical bars standard deviation. Significant differences are indicated by * *p* < 0.05, ** *p* < 0.01, *** *p* < 0.001, Mann–Whitney U test.

**Figure 4 microorganisms-09-01492-f004:**
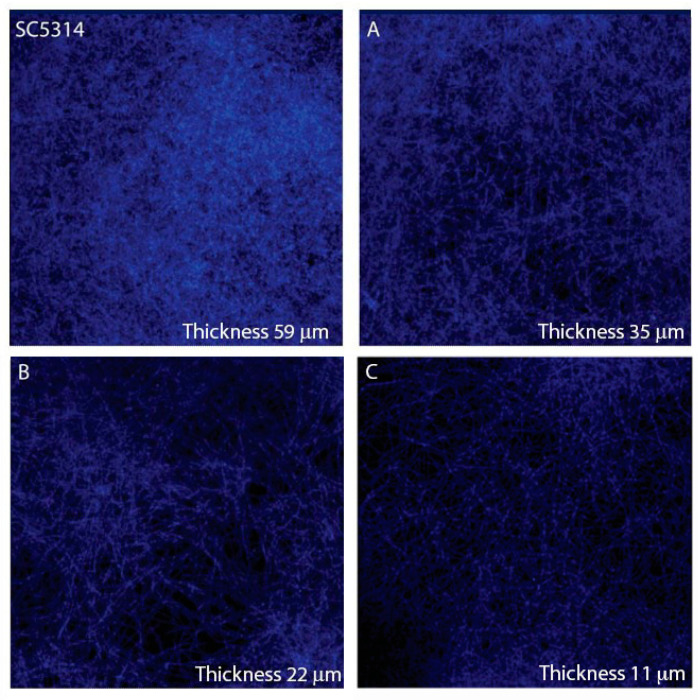
Visualization of *C. albicans* SC5314 biofilms in the presence of (**A**) 4-HBA (3.5 μM,) (**B**) VAL (245 μM), (**C**) and Anthocran^®^ (0.1 mg/mL), compared to control. Thickness was measured by z-scanning using the instrument software.

**Figure 5 microorganisms-09-01492-f005:**
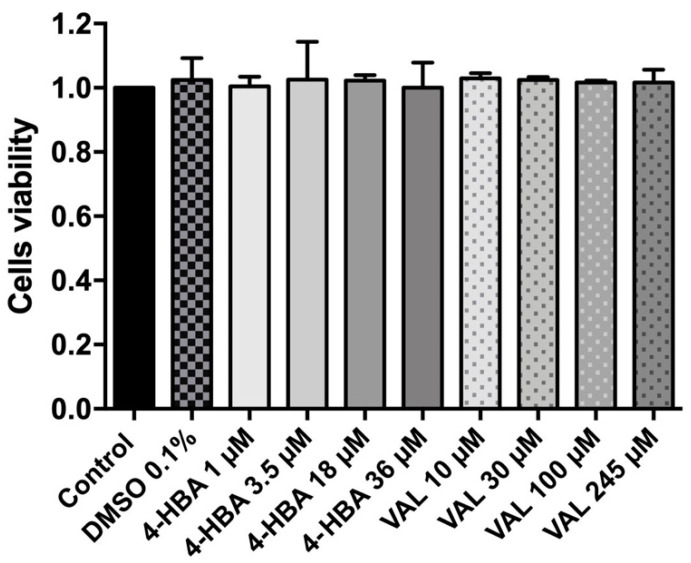
4-HBA and VAL cytotoxicity assay. Data represent the mean ± SD of three different experiments, performed in quadruplicates, compared by one-way ANOVA. Control: untreated cells; vehicle: DMSO 0.1%.

**Figure 6 microorganisms-09-01492-f006:**
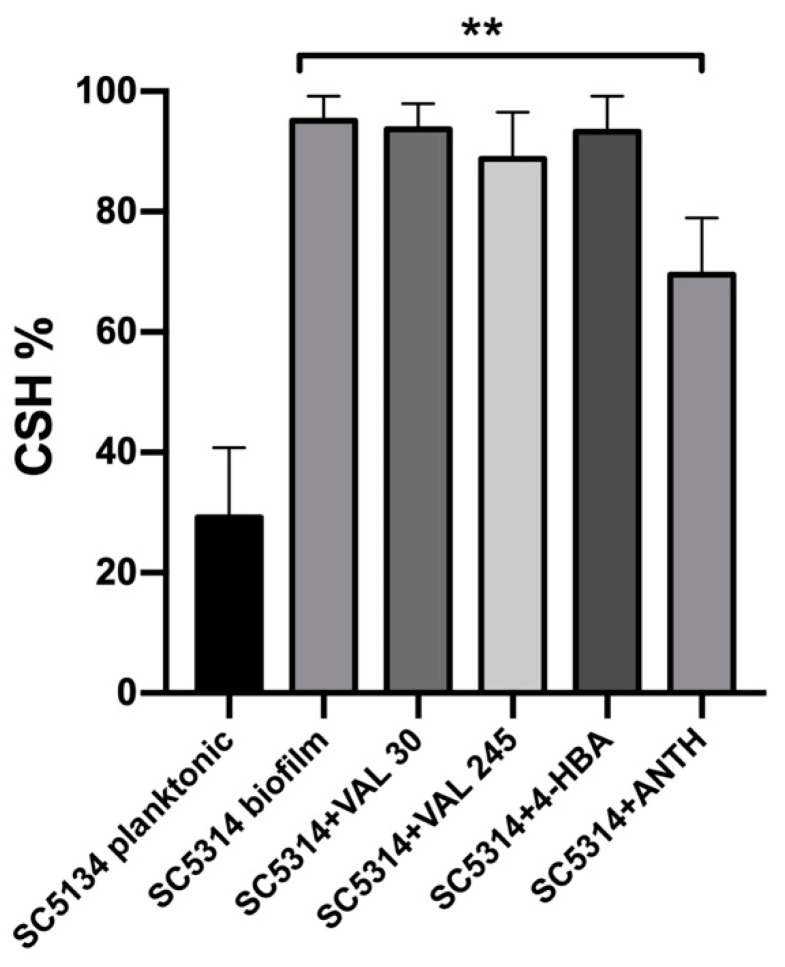
Cell surface hydrophobicity (CSH) is not altered by 4-HBA (3.5 μM) and VAL (30 and 245 μM) alone, whereas Anthocran^®^ 0.1 mg/mL significantly affects the yeast surface hydrophobicity. Data represented the mean ± SD of three independent experiments. Data sets were compared by one-way ANOVA, ** *p* < 0.01.

**Figure 7 microorganisms-09-01492-f007:**
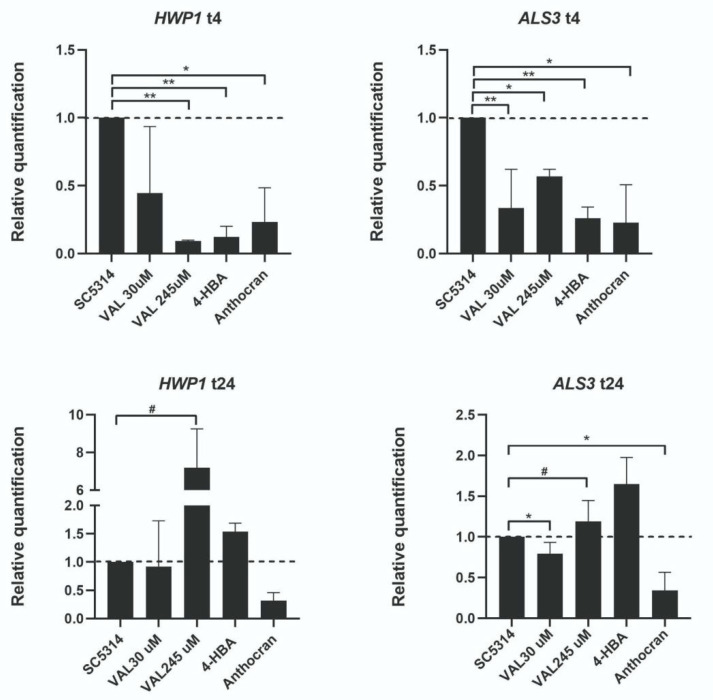
Modulation of *C. albicans* gene expression by cranberry metabolites. *C. albicans* SC5314 was induced to form biofilm in the presence/absence of two concentrations of VAL (30 μM and 245 μM) and of 4-HBA (3.5 μM). At 4 h, both *HWP1* and *ALS3* genes were significantly downregulated by either VAL or 4-HBA (* *p* < 0.05; ** *p* < 0.01). At 24 h, the modulatory effect was lost and, especially with VAL 245 μM a paradoxical effect, i.e., upregulation, was observed (# *p* < 0.05). Data represent the mean ± SD of three different experiments performed in triplicates.

**Table 1 microorganisms-09-01492-t001:** Clinical isolates used in the study and biofilm-forming ability.

ID Strain	Species	Source	Biofilm Production *
SC5314	*C. albicans*	Reference strain	+++
g11	*C. albicans*	Vaginal swab	+++
g23	*C. albicans*	Vaginal swab	++
g29	*C. albicans*	Vaginal swab	+++
g35	*C. albicans*	Vaginal swab	++
g44	*C. albicans*	Vaginal swab	+++
g49	*C. albicans*	Vaginal swab	+/−
g67	*C. albicans*	Vaginal swab	++
g69	*C. albicans*	Urinary tract infection	+++
g14	*C. albicans*	Urinary tract infection	+++
g53	*C. albicans*	Urinary tract infection	+++

* high (+++), medium (++) and low (+) biofilm producers were determined according to the following crystal violet absorbance values at 540 nm: +++ > 1.17; ++ = 0.44 – 1.17; +/− < 0.44 [14].

## Data Availability

The study was registered at www.isrctn.org as ISRCTN32556347.

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
