# Peer review of "Candida albicans Biofilm Inhibition by Two Vaccinium macrocarpon (Cranberry) Urinary Metabolites: 5-(3′,4′-DihydroxyPhenyl)-γ-Valerolactone and 4-Hydroxybenzoic Acid"

_microorganisms, 2021, doi:10.3390/microorganisms9071492_

Round 1

Reviewer 1 Report

This article is a complex one and addresses a topical issue.

I still have some questions and comments on this article.

  1. The title of the article is not the most appropriate. The title assumes that it is a single compound (5- (3 ′, 4′-dihydroxyphenyl) -γ-valerolactone). In the abstract, however, 2 compounds are already presented and in the article is presented the evaluation of several metabolites of cranberry extract. I would suggest changing the title to be consistent with the content of the article.
  2. In figure 1 what exactly is the term Candida isolates? Table 1 shows 10 clinically isolated strains of C. albicans. They are denoted by g11-g53. To which of these does the term Candida isolates in Figure 1 refer? Because the explanation of the figure is that the black bar represents ten C. albicans clinically isolated biofilms. Could you be more specific?
  3. In material and method the authors propose For 4-HBA and VAL, various concentrations (3.5-36 μM and 10-245 μM, respectively) were assessed. For the other compounds 10 μM concentration was used. However, in paragraph 2.3 the authors propose the following Mass spectrometry (MS) qualitative analysis, based on a target approach, highlighted several increased metabolites in active urine fractions [11]. We thus tested the direct effect of these metabolites (10 μM) against C. albicans SC5314 biofilm. So in this case was the 10 μM concentration also used for 4-HBA? this concentration does not appear later. Was VAL tested at 10 μM on C. albicans strain SC5314? This compound appears only when tested on isolated strains.
  4. How were the concentrations tested for VAL chosen? The authors propose In particular, the most active fraction (U-12h) was characterized by the presence of high concentrations (245.24 ± 143.59 μM) of 5- (3 ′, 4′-dihydroxyphenyl) -γ-valerolactone (VAL). This compound was tested on clinically isolated strains at concentrations of 10, 30 and 245 micromoles. Why not concentrations of 50 or 100?
  5. In material and method the authors propose To investigate the morphological and structural changes induced by Anthocran® and its metabolites on hyphal growth and biofilm, C. albicans reference strain SC5314 was used for CLSM evaluation after 24 h exposure to 30 μM or 245 μM of VAL and 0.1 mg/ml of Anthocran®. In the results section, however Figure 4. Shows Visualization of C. albicans SC5314 biofilms in the presence of (A) 4-HBA (3.5 μM,) and (B) VAL (245 μM), compared to control. Can you explain the discrepancy?

Author Response

POINT-BY-POINT REPLY

This article is a complex one and addresses a topical issue. I still have some questions and comments on this article.

Q1- The title of the article is not the most appropriate. The title assumes that it is a single compound (5- (3 ′, 4′-dihydroxyphenyl) -γ-valerolactone). In the abstract, however, 2 compounds are already presented and in the article is presented the evaluation of several metabolites of cranberry extract. I would suggest changing the title to be consistent with the content of the article.

R1- We thank the reviewer for this suggestion. We amended the title in “Candida albicans biofilm inhibition by two Vaccinium macrocarpon (cranberry) urinary metabolites: 5-(3′,4′-dihydroxyphenyl)-γ-valerolactone and 4-hydroxybenzoic acid”, taking in consideration also reviewer#2 comment about the latin name for cranberry to be highlighted in the title.

Q2- In figure 1 what exactly is the term Candida isolates? Table 1 shows 10 clinically isolated strains of C. albicans. They are denoted by g11-g53. To which of these does the term Candida isolates in Figure 1 refer? Because the explanation of the figure is that the black bar represents ten C. albicans clinically isolated biofilms. Could you be more specific?

R2- We thank the reviewer for highlighting this missing information. In figure 1, graph bars are representative of all clinical isolate absorbances (mean ± SD). This aspect has been highlighted in the figure caption.

Q3- In material and method the authors propose For 4-HBA and VAL, various concentrations (3.5-36 μM and 10-245 μM, respectively) were assessed. For the other compounds 10 μM concentration was used. However, in paragraph 2.3 the authors propose the following Mass spectrometry (MS) qualitative analysis, based on a target approach, highlighted several increased metabolites in active urine fractions [11]. We thus tested the direct effect of these metabolites (10 μM) against C. albicans SC5314 biofilm. So in this case was the 10 μM concentration also used for 4-HBA? this concentration does not appear later. Was VAL tested at 10 μM on C. albicans strain SC5314? This compound appears only when tested on isolated strains.

R3- The initial concentration of all compounds was 10 μM (including 4-HBA). Indeed, the MS targeted analysis was a qualitative measurement of analytes. We also did test 10 μM VAL on C. albicans SC5314 reference strain and we observed a strong reduction on biofilm biomass (about 70%). However, clinical isolates were responding to this concentration in a variable manner (as shown in Figure 3), as most of them resulted unaffected. We did not include VAL in Figure 2, based on MS target approach, for consistency, but we did report the 10 μM data for clinical isolates in the Figure 3.

Q4- How were the concentrations tested for VAL chosen? The authors propose In particular, the most active fraction (U-12h) was characterized by the presence of high concentrations (245.24 ± 143.59 μM) of 5- (3 ′, 4′-dihydroxyphenyl) -γ-valerolactone (VAL). This compound was tested on clinically isolated strains at concentrations of 10, 30 and 245 micromoles. Why not concentrations of 50 or 100?

R4-VAL10 μM was chosen for analogy with the concentration used for targeted analysis; 30 μM was the average of VAL concentration in non-active urinary fractions. We added this approach in the text.

Q5- In material and method the authors propose To investigate the morphological and structural changes induced by Anthocran® and its metabolites on hyphal growth and biofilm, C. albicans reference strain SC5314 was used for CLSM evaluation after 24 h exposure to 30 μM or 245 μM of VAL and 0.1 mg/ml of Anthocran®. In the results section, however Figure 4. Shows Visualization of C. albicans SC5314 biofilms in the presence of (A) 4-HBA (3.5 μM,) and (B) VAL (245 μM), compared to control. Can you explain the discrepancy?

R5- We thank the reviewer for pointing out this discrepancy. We have now added the missing CLSM image (Anthocran® 0.1 mg/ml- treated biofilm) to the panel.

Reviewer 2 Report

The manuscript #microorganisms-1274577, entitled “Candida albicans biofilm inhibition by 5-(3′,4′-dihydroxyphenyl)-γ-valerolactone, a metabolite of cranberry” by Ottaviano et al. presents the anti-adhesive and anti-biofilm activities of urinary metabolites of cranberries against C. albicans fungus. The presented paper is very well written, and brings novelty to the field. The design of the study is logical and well planned. The fact that the Authors have used clinical isolates, which were collected from human urinary tract to investigate the effect of urinal metabolites is one of the examples of logical design of the study. Most of my comments concern minor (mostly editorial) issues.

Minor issues

Please check the Authors names. Those presented in the manuscript do not match the names in the submission system (Jess versus Jessica etc.).

In my opinion, the latin name for cranberry should be used in the title of the manuscript.

In the Introduction section, the Authors should highlight the fact, that the investigated compounds are urinary metabolites of cranberries. It is not clear when reading the Introduction.

The usage of Latin names for the microorganisms can be improved. When the Authors mention the fungi for the first time, they should use the full name, e.g. Candida albicans. However, every other usage of the name should be shortened into C. albicans. This must be corrected throught the manuscript.

The authors should unify how they present liters. Sometimes they write “ml”, and sometimes “mL” – it should be unified. Both “l” and “L” are correct, but use only one way of writing it. Similarly “hours” should be unified (sometimes it is written as “24 h”, and sometimes as “1 hour”). Also, the number should be divided by space (eg. 24 h, not “24h”). Similarly, when presenting p value, sometimes the Authors missed spaces between the symbols. Proteins in case of Candida albicans should be written as “Als1p” not “ALS1p”. Those editorial issues should be corrected throughout the manuscript.

When presenting methods 2.4 and 2.6 the equations should be presented according to the journal guidelines.

Table 1: In the caption please elaborate that the Authors have used the criteria established by Marcos-Zambrano et al.

Figure 1: The authors should elaborate on the effect of the urinary fractions, from patients not receiving cranberry extract.

Figure 3: 4-HBA at concentration equal 3.5 uM inhibited biofilm formation. However, in the figure the Authors presented 18 uM, which had no effect. This should be discussed.

In the Discussion section the Authors have written: “Indeed, Candida spp. are responsible for a high percentage of catheter-related UTIs within the nosocomial setting that, if not efficiently treated, can eventually lead to bloodstream invasion [24].”. Have the authors considered performing anti-biofilm studies using catheters ? Similarly as presented in: Janek, Łukaszewicz, Krasowska. BMC Microbiology 2012, 12:24.

In the Discussion, the authors stated that: “for other Candida species which biofilm lacks the presence of pseudohyphae and/or hyphae”. In my opinion the Authors should provide examples, such as C. glabrata.

Author Response

POINT-BY-REPLY

The manuscript #microorganisms-1274577, entitled “Candida albicans biofilm inhibition by 5-(3′,4′-dihydroxyphenyl)-γ-valerolactone, a metabolite of cranberry” by Ottaviano et al. presents the anti-adhesive and anti-biofilm activities of urinary metabolites of cranberries against C. albicans fungus. The presented paper is very well written, and brings novelty to the field. The design of the study is logical and well planned. The fact that the Authors have used clinical isolates, which were collected from human urinary tract to investigate the effect of urinal metabolites is one of the examples of logical design of the study. Most of my comments concern minor (mostly editorial) issues.

Minor issues

Q1- Please check the Authors names. Those presented in the manuscript do not match the names in the submission system (Jess versus Jessica etc.).

R1- We thank the reviewer for pointing out this typo. We carefully checked the authors’ names and amended them when necessary.

Q2- In my opinion, the latin name for cranberry should be used in the title of the manuscript.

R2- According to both reviewers’ comments, we changed the title in “Candida albicans biofilm inhibition by two Vaccinium macrocarpon (cranberry) urinary metabolites: 5-(3′,4′-dihydroxyphenyl)-γ-valerolactone and 4-hydroxybenzoic acid”

Q3- In the Introduction section, the Authors should highlight the fact, that the investigated compounds are urinary metabolites of cranberries. It is not clear when reading the Introduction.

R3- We thank the reviewer for highlighting the missing information. We amended the title specifying the two cranberry metabolites that are the focus of our work. Moreover, in the introduction, we added in the scope description the 4-hydroxybenzoic acid.

Q4- The usage of Latin names for the microorganisms can be improved. When the Authors mention the fungi for the first time, they should use the full name, e.g. Candida albicans. However, every other usage of the name should be shortened into C. albicans. This must be corrected throught the manuscript.

The authors should unify how they present liters. Sometimes they write “ml”, and sometimes “mL” – it should be unified. Both “l” and “L” are correct, but use only one way of writing it. Similarly “hours” should be unified (sometimes it is written as “24 h”, and sometimes as “1 hour”). Also, the number should be divided by space

(eg. 24 h, not “24h”). Similarly, when presenting value, sometimes the Authors missed spaces between the symbols. Proteins in case of Candida albicans should be written as “Als1p” not “ALS1p”. Those editorial issues should be corrected throughout the manuscript.

R4- We thank the reviewer for the careful revision. We have now harmonized all the mentioned words/acronym throughout the text.

Q5- When presenting methods 2.4 and 2.6 the equations should be presented according to the journal guidelines.

R5- We revised the equations by using the Word function as mentioned in the journal guidelines

Q6-Table 1: In the caption please elaborate that the Authors have used the criteria established by Marcos-Zambrano et al.

R6- We thank the reviewer for his/her suggestions. We have added the applied criteria at the bottom of the table.

Q7- Figure 1: The authors should elaborate on the effect of the urinary fractions, from patients not receiving cranberry extract.

R7- We did check urinary fractions collected before Anthocran® supplementation for inhibitory effect with no significant results. We did report this data in the first paper. To clarify this aspect, we have added a sentence when discussing these results

Q8- Figure 3: 4-HBA at concentration equal 3.5 uM inhibited biofilm formation. However, in the figure the Authors presented 18 uM, which had no effect. This should be discussed.

R8- We thank the reviewer for this suggestion. We added a sentence in the result section about the strain-specific behaviour observed when treating Candida biofilm with 4HBA concentrations higher than the one recovered in the most active urinary fraction. Indeed, we did observe a bimodal response, with some strains being unaffected by the increased concentration. We can speculate, as observed for other antifungal compounds, that a paradoxical growth of some strains that remain susceptible at low or intermediate concentration.

Q9- In the Discussion section the Authors have written: “Indeed, Candida spp. are responsible for a high percentage of catheter-related UTIs within the nosocomial setting that, if not efficiently treated, can eventually lead to bloodstream invasion [24].”. Have the authors considered performing anti-biofilm studies using catheters ? Similarly as presented in: Janek, Łukaszewicz, Krasowska. BMC Microbiology 2012, 12:24.

R9- We agree with the reviewer’s observation about the complementary strategy of using functionalized biomaterials with antiadhesive properties. We are also working on this field (Mauri E, Naso D, Rossetti A, Borghi E, Ottaviano E, Griffini G, Masi M, Sacchetti A, Rossi F. Design of polymer-based antimicrobial hydrogels through physico-chemical transition. Mater Sci Eng C Mater Biol Appl. 2019). We have added a sentence for discussing this matter.

Q10- In the Discussion, the authors stated that: “for other Candida species which biofilm lacks the presence of pseudohyphae and/or hyphae”. In my opinion the Authors should provide examples, such as C. glabrata.

R10- Thanks for the suggestion. We have modified the manuscript accordingly.

Round 2

Reviewer 1 Report

I appreciate the authors' response as well as the changes and additions made in the article.